# Fuel Type Mapping Using a CNN-Based Remote Sensing Approach: A Case Study in Sardinia

**Andrea Carbone** [1,*], **Dario Spiller** [2] and **Giovanni Laneve** [2]

1    Department of Civil, Constructional and Environmental Engineering (DICEA), Sapienza University of Rome, Via Eudossiana 18, 00184 Rome, Italy

2    School of Aerospace Engineering, Sapienza University of Rome, Via Salaria 851, 00138 Rome, Italy; dario.spiller@uniroma1.it (D.S.); giovanni.laneve@uniroma1.it (G.L.)

*    Correspondence: and.carbone@uniroma1.it

**Abstract:** Accurate fuel mapping is crucial for effectively determining wildfire risk and implementing management strategies. The primary challenge in fuel type mapping lies in the need to develop accurate and efficient methods for identifying and categorizing the various combustible materials present in an area, often on a large scale. In response to this need, this paper presents a comprehensive approach that combines remote sensing data and Convolutional Neural Network (CNN) to discriminate between fire behavior fuel models. In particular, a CNN-based classification approach that leverages Sentinel-2 imagery is exploited to accurately classify fuel types into seven preliminary main classes (broadleaf, conifers, shrubs, grass, bare soil, urban areas, and water bodies). To further refine the fuel mapping results, subclasses were generated from the seven principles by using biomass and bioclimatic maps. These additional maps provide complementary information about vegetation density and climatic conditions, respectively. By incorporating this information, we align our fuel type classification with the widely used Standard Scott and Burgan (2005) fuel classification system. The results are highly promising, showcasing excellent CNN training performance with all three metrics—accuracy, recall, and F1 score—achieving an impressive 0.99%. Notably, the network exhibits exceptional accuracy in a test case conducted in the southern region of Sardinia, successfully identifying Burnable classes in previously unseen pixels: broadleaf at 0.99%, conifer at 0.79%, shrub at 0.76%, and grass at 0.84%. The proposed approach presents a valuable tool for enhancing fire management, contributing to more effective wildfire prevention and mitigation efforts. Thus, this tool could be leveraged by fire management agencies, policymakers, and researchers to improve the determination of wildfire risk and management.

**Keywords:** fire management; fuel mapping; Sentinel 2; CNN classification; Scott and Burgan fuel classification system



## 1. Introduction

Accurate and updated information on fuel types and distribution is essential for effective wildfire management [1–3], ecosystem planning [4], and natural resource management [5,6]. Fuel maps, which provide detailed spatial information about vegetation fuel characteristics, play a crucial role in these domains. Traditionally, fuel mapping has relied on ground-based field surveys and visual interpretation of aerial imagery. However, these methods are time-consuming, costly, and limited in coverage. In recent years, remote sensing technology has brought about significant advancements in addressing these limitations. Multispectral and hyperspectral imagery, capable of capturing a wide range of spectral information from the Earth's surface, have emerged as powerful tools to enhance fuel mapping and related endeavors. Indeed, this spectral information can be used to discriminate between different fuel types based on their unique spectral signatures [7–9]. Additionally, machine learning algorithms, particularly those based on deep learning techniques, have

demonstrated exceptional capabilities in extracting complex patterns and relationships from large-scale remote sensing datasets. The combination of remote sensing and machine learning offers numerous advantages for classification tasks providing valuable ancillary information about Earth's surface proprieties [10–13]. It enables the generation of fuel maps over large areas in a timely and cost-effective manner, surpassing the limitations of traditional field-based surveys. For instance, I. Chrysafis et al. [14] achieved enhanced accuracy in fuel type classification by developing random forest classification models. Their approach involved combining data from both passive and active sensors sourced from the Sentinel family of satellites, in conjunction with topographic variables. This methodology was applied to map fuel types in northeastern Greece. Additionally, Ensley-Field et al. [15] developed a fuel model that considers the fuel load from the previous year and utilizes productivity estimates derived from early spring remotely sensed data to predict fuel load at specific locations. D'Este et al. [16] conducted a study on the estimation of fine dead fuel load. They utilized field data, multi-source remote sensing data, and machine learning techniques (Random Forest (RF) and Support Vector Machine (SVM)) to support decision-making and regional wildfire risk management. The results showed that Random Forest performed better in their analysis. Again, the study of Santos et al. [17] aims to improve the vegetation representation of the fuel load and moisture content in Southern Portugal. Field samples and satellite data from Sentinel-2 were used with the RF classifier for analysis. In recent years, Aragoneses E. and Chuvieco E. [18] developed a methodology for fuel mapping using Sentinel-3 images, vegetation continuity, and biomass data. They compared the SVM and RF algorithms and found that SVM performed better in their cases. In [19], R.U. Shaik et al. employed an automatic semi-supervised SVM approach to distinguish between 18 different types of wildfire fuel using PRISMA hyperspectral imagery, resulting in an overall accuracy of 87%. The study proposed by Maniatis Y. et al. [20] developed a fire risk estimation model by using an SVM algorithm, validated by wildfires in 2020 and 2021. Again, an SVM-based method is exploited by Garcia M. et al. [21] to classify fuel types by using multispectral data and vertical information provided by the LiDAR, reaching an overall accuracy of around 88%. In the work of Alipour M. et al. [22], a combined approach of CNN and deep neural network is developed to provide accurate large-scale fuel estimation. The study specifically explores the utilization of Convolutional Neural Networks for fuel mapping. Furthermore, it is worth noting that CNNs have shown promising results in the context of wildfire detection [23–32] and vegetation mapping [33]. Thus, the promising results achieved through the integration of remote sensing and machine learning techniques provide strong motivation for continued exploration and advancement in the field of fuel mapping. In this paper, the area of interest is Sardinia, which has gained notoriety for the frequent prevalence of wildfires. Over the preceding decade, this island has witnessed an annual average of 1008 fire incidents, constituting a significant 20% of the overall national tally [34]. Several studies have already been conducted to generate fuel maps in Sardinia; in [35], Bajocco et al. created a phenological fuel map for fire-prone Sardinia using MODIS NDVI data from 2000 to 2012. It segments vegetation based on seasonality, clusters phenological units, and assesses their fire risk. The findings suggest that satellite data can accurately predict fire risk, offering a basis for fire distribution models and biogeographic studies. Again, in [36], Oliveria et al. develop a framework to assess wildfire vulnerability in Mediterranean Europe, using data such as population density, fuel types, and protected areas to create multidimensional maps. It was applied in various regions, with validation showing over 72% accuracy. In [37], Salis et al. assess fine-scale wildfire hazard and exposure. Using the Minimum Travel Time algorithm and historical data, the study identifies hot-spot areas and enhances regional understanding of wildfire dynamics. In [38], Aragonese et al. introduce a European fuel classification system with 85 types in six categories, crucial for fire risk management. It explains the mapping process at a 1 km resolution, achieving an 88% accuracy for primary fuel types.

In this paper, a novel method that combines remote sensing data and CNNs to discriminate between fire behavior fuel models is presented. Specifically, we leverage Sentinel-2

imagery and a CNN-based classification approach to accurately classify fuel types into seven preliminary main classes: conifers, broadleaf, shrubs, grass, bare soil, urban areas, and water bodies. By training the CNN on a large dataset of annotated Sentinel-2 imagery, the model can learn complex patterns and relationships between spectral signatures and fuel types. This enables precise discrimination between different fuel types, facilitating more effective wildfire risk assessment and management. To refine the fuel mapping results, further subclasses from the seven main classes are generated using Above-Ground Biomass (AGB) and Bioclimatic (BC) maps. These additional maps provide valuable information about vegetation density and moisture conditions, respectively. By incorporating this information, the fuel type classification is aligned with the widely used Standard Scott and Burgan fuel classification system [39]. This refinement step allows for a more detailed and comprehensive assessment of fuel types, enhancing the accuracy and effectiveness of fire detection and management efforts. The proposed approach has the potential to support fire management agencies, policymakers, and researchers in their endeavors to enhance fire prevention and mitigation, ultimately minimizing the impact of wildfires on ecosystems and human populations. The latest advancements in fuel map generation have been the focus of this research, demonstrating the effectiveness of CNN-based deep learning techniques in generating precise fuel maps.

This paper is organized as follows. In Section 2, the dataset is described in terms of areas of interest, multispectral Sentinel-2 imagery, ancillary maps (ABG and BC), and the Standard Scott and Burgan fuel classification system. A description of the methodology is provided in Section 3, where the CNN-based fuel map classification is described. The results of the training of performance and accuracy are reported in Section 4, while Section 5 deals with the discussion of the numerical findings. Finally, concluding remarks are provided in Section 6.

## 2. Dataset Definition

This section includes a comprehensive exploration of the designated area of interest, an introduction of the Sentinel-2 multispectral data, an in-depth analysis of the AGB and BC maps, and a presentation of the Standard Scott and Burgan fuel type.

### 2.1. Area of Interest

As shown in Figure 1, the focus of this study centers around Sardinia, an island positioned in the southern region of Italy and recognized as the second-largest landmass in the Mediterranean Sea. The area is characterized by gusty winds, intermittently wet winters, and scorching, sun-drenched summers. The temperature fluctuations are noteworthy, ranging from 10 °C during the winter months of January and February to a balmy 24–25 °C during the summer period encompassing July and August [40]. To assess the fuel map under high-risk fire probability conditions and during periods of significantly increased fuel load, a satellite image captured by the Sentinel-2 on 17 July 2022, during the summer season, was selected for analysis. The particular area selected and situated east of Cagliari consists of an expansive forested landscape that traverses undulating terrain, reaching altitudes of approximately 800 m, encompassing a total land area of 32 km$^2$.

The CNN training dataset was obtained through a visual inspection of the pseudo-color image derived from the Sentinel-2 image of the region of interest. The selection of each class was carried out manually with the assistance of the four land cover maps shown in Figure 2. This set of four maps was employed as ground truth, each distinctly representing specific classes: (a) Land Cover (LC) 2021, provided by ISPRA ("Istituto Superiore per la Protezione e la Ricerca Ambientale"), represents the bio-physical coverage of the Earth's surface, including the Broadleaf, Conifer, Shrub, and Grass classes (https://www.isprambiente.gov.it/it, accessed on 10 September 2023); (b) ESA World Cover (WC) 2021 identifies seven classes representing different land surface types, including Tree Cover, Shrubland, Grassland, and Cropland (https://worldcover2021.esa.int, accessed on 10 September 2023); (c) the Forest Type product (FTY) is a part of the European

Environment Agency (EEA) Copernicus Land Monitoring Service, and it provides a forest classification with three thematic classes: non-tree areas, broadleaved forest, and coniferous forest (https://land.copernicus.eu/pan-european/high-resolution-layers/forests/forest-type-1/status-maps/forest-type-2018, accessed on 10 September 2023); (d) Grassland (GRA) 2018 is also developed under the EEA Copernicus Land Monitoring Service, and it offers a basic land cover classification with two thematic classes: grass and no grass (https://land.copernicus.eu/pan-european/high-resolution-layers/grassland/status-maps/grassland-2018, accessed on 10 September 2023). Through meticulous cross-referencing and diligent visual inspections of these maps, the training dataset was carefully selected. The resulting dataset is reported in Table 1, which shows the number of labeled pixels for each class.

**Table 1.** Number of labeled reference pixels for CNN training dataset.

| Broadleaf | Conifer | Shrub | Grass | Bare Soil | Urban | Water |
|---|---|---|---|---|---|---|
| 18,093 | 2890 | 2924 | 1242 | 112 | 14,309 | 91,452 |

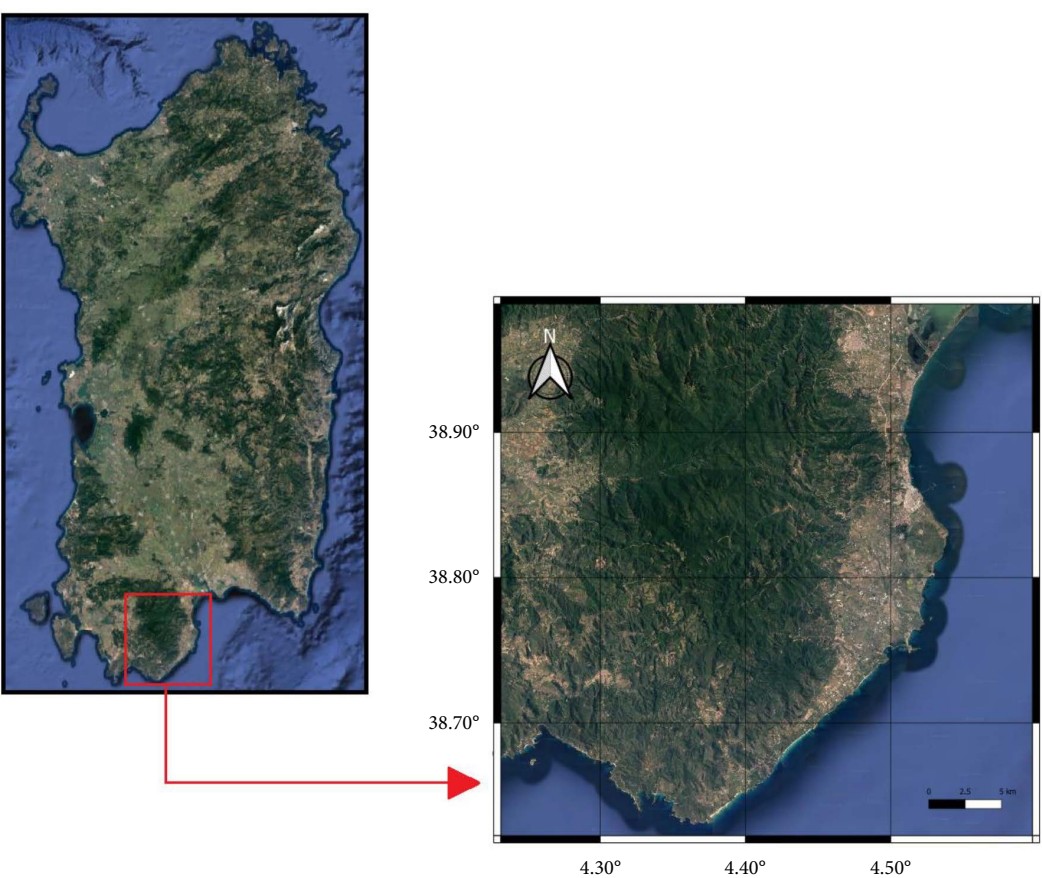

**Figure 1.** Geolocation of area of interest (south of Sardinia).

One can observe that the higher frequency of the Water class compared to the others can be attributed to its ease of identification and widespread presence within the examined image. These favorable conditions facilitated the labeling process by allowing for a larger number of instances to be accurately labeled. Conversely, the remaining classes exhibited increasingly intricate discrimination challenges. Despite their substantial representations in the image, a more extensive and demanding exploration was required, resulting in smaller characteristic classes. This problem was particularly pronounced for the Bare Grounds and Grass classes, where the combination of a limited amount of samples and the inherent complexity of reducing their noise led to a deliberate preference for a smaller

sample set. These specific classes often showed a propensity to coexist with other land cover categories within the study area. For example, the class "Grass" was often associated with shrubs and agricultural fields, necessitating the identification of well-manicured gardens, although such cases proved relatively rare. In contrast, the class "Bare soil" showed a limited presence within the study region and, when identified, often showed contamination by grass or shrubs. In summary, the identification and isolation of these classes posed a significant challenge because of their recurrent overlaps with other land cover categories within the geographic boundaries of our study. However, these samples remained unequivocally representative of the reference class despite their small numbers.

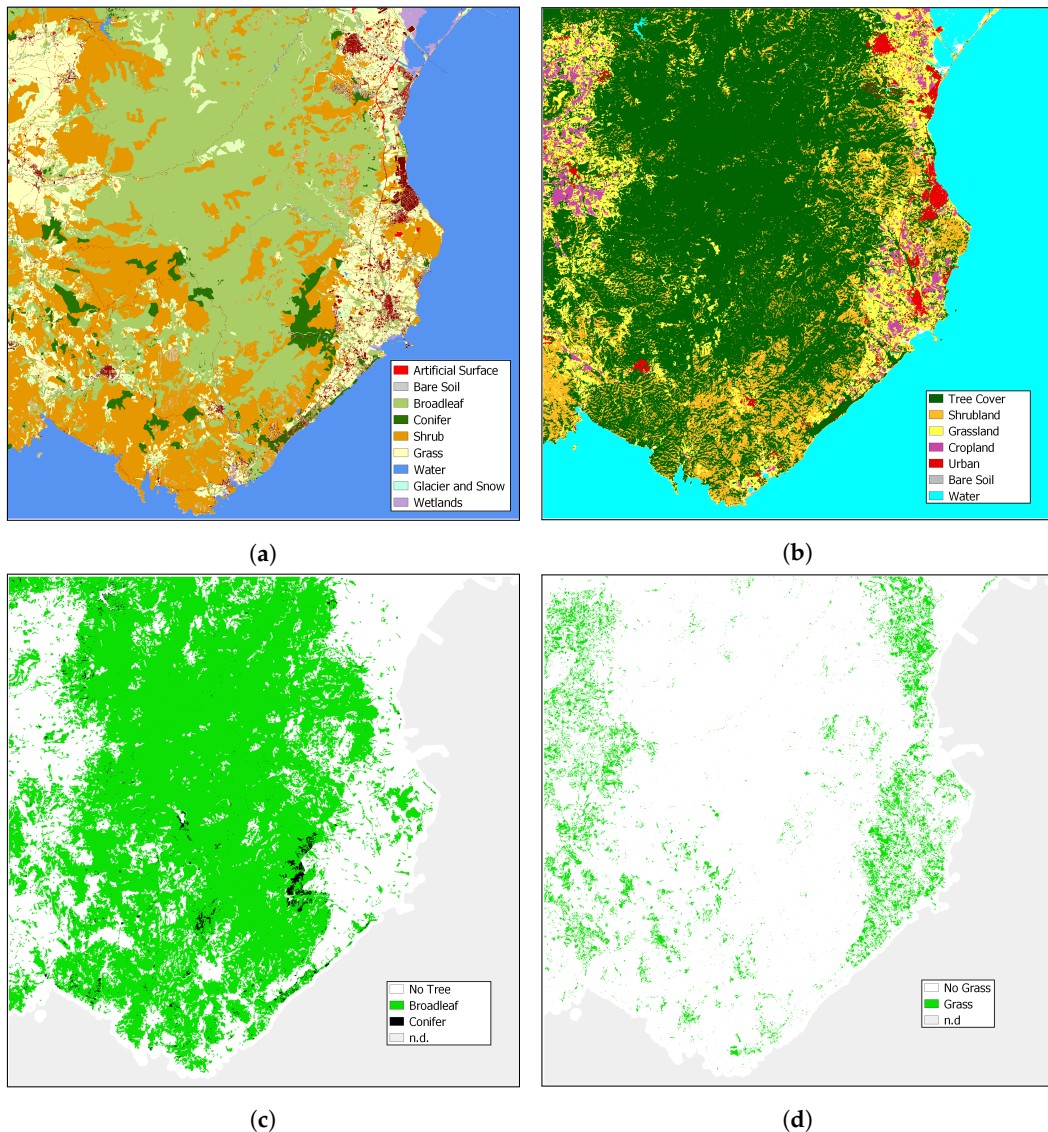

**Figure 2.** Reference maps used both to collect the dataset for training and to evaluate the performance of the trained CNN. (**a**) Land Cover provided by ISPRA; (**b**) World Cover provided by ESA; (**c**) Forest Type provided by Copernicus Project; (**d**) Grass Cover provided by Copernicus Project.

### 2.2. Sentinel Data

This section provides a description of the Sentinel multispectral data used for training and validating the CNN model. In our study, spectral images from Sentinel-2 acquired on 17 July 2022 were utilized as the primary dataset for training the CNN. The Sentinel-2 satellite sensor captures images in multiple spectral bands, which provide valuable information about Earth's surface. The Sentinel-2 mission, launched by the European Space Agency (ESA), aims to provide global and systematic observations of the Earth's

land and coastal areas. It consists of two identical satellites, Sentinel-2A and Sentinel-2B, which together cover the entire Earth's landmass every 5 days. The satellites acquire high-resolution imagery in 13 spectral bands with spatial resolutions of 10, 20, and 60 m, enabling the monitoring of land cover, vegetation dynamics, and environmental changes. For our analysis, all of the sensor bands were resized to a spatial resolution of 10 m. To further improve classification, vegetation indices were calculated from the Sentinel-2 bands and added to the spectrum dataset: the Normalized Difference Vegetation Index (NDVI), Enhanced Vegetation Index (EVI), and Normalized Difference Water Index (NDWI), which are reported in Table 2. These are widely used to assess vegetation health and density [41–45].

**Table 2.** Index referred to Sentinel-2.

| Index | Description |
|:---:|:---:|
| $NDVI = \dfrac{B08 - B04}{B08 + B04}$ | Range between $-1$ (non-vegetated surfaces such as bare soil or water) and $+1$ (high density of healthy vegetation) |
| $EVI = \dfrac{2.5 \cdot (B08 - B04)}{B08 + 6 \cdot B04 - 7.5 \cdot B02 + 1.0}$ | Range between $-1$ (non-vegetated surfaces or stressed vegetation) and $+1$ (high density and healthier vegetation) |
| $NDWI = \dfrac{B03 - B08}{B03 + B08}$ | Range between $-1$ (non-water surfaces) and $+1$ (high likelihood of water presence) |

*2.3. Ancillary Maps*

This section provides a detailed overview and description of how the AGB and BC ancillary maps are carried out, highlighting their crucial contribution to enhancing the classification process. Indeed, our preliminary land cover classification relies on a limited number of classes, which may not capture the fine-scale variability within a given region. Thus, to address this limitation, an approach that utilizes additional data layers is exploited, specifically AGB and BC maps, to generate a more detailed subclassification scheme from a primary land cover classification obtained through CNN. Then, the final subclassification is aligned with the widely accepted Scott and Burgan fuel classification system.

- The *Above-Ground Biomass Map* represents the total mass of living vegetation per unit area, typically expressed in metric tons per hectare (unit: tons/ha). The AGB map used in this study was obtained from the European Space Agency's (ESA's) Climate Change Initiative (CCI) program [46], which is an ESA project to provide long-term, high-quality climate data records to support climate change research and related applications (the AGB map can be downloaded for free at this link: https://data.ceda.ac.uk/neodc/esacci/biomass/data/agb/maps, accessed on 25 May 2023). The AGB map of ESA exhibits a continuous range of values; however, in this work, the biomass values are normalized as percentages relative to the maximum value. Subsequently, the AGB map is divided into three macrogroups based on the following percentage thresholds: the first group includes all values below 40%, the second group encompasses values between 40% and 70%, and the third group includes values exceeding 70%. This is performed to facilitate analysis and align with the Scott and Burgan fuel type classification. Indeed, this approach allowed us to establish a direct correlation between the biomass percentages and the Scott and Burgan fuel type classification, specifically the distinction between low, medium, and high forest density. Furthermore, due to the initial resolution of the raster being 100 m, it was imperative to rescale the map to a finer resolution of 10 m. This resampling process was performed using QGIS software, ensuring the preservation of relevant spatial information and maintaining data integrity throughout the analysis. The post-processed AGB map is

shown on the left side of Figure 3, where the three main classes mentioned above are reported. As expected from the RGB image in Figure 1, a predominant representation of classes below the 70 percent threshold is observed, indicating a dense vegetation cover exclusively in the most remote regions of the study area, specifically the peaks located within the inner mountains. On the contrary, the remaining portion of the region of interest predominantly consists of agricultural fields, grasses, or sparsely forested areas. This finding emphasizes the crucial importance of incorporating the use of this map, as it clearly reveals the pronounced disparity in land cover composition and illuminates the unique ecological features present in the study area.

- The *Climate Map* is derived from the BC map of Sardinia that was developed through a collaboration among several institutions:

  - ARPAS—the Regional Agency for Environmental Protection of Sardinia (Agenzia Regionale per la Protezione dell'Ambiente della Sardegna)—Meteoclimatic Department, Sassari: ARPAS is the regional agency responsible for environmental protection in Sardinia. Their Meteoclimatic Department contributed to data collection.

  - University of Sassari, Department of Natural and Territorial Sciences, Sassari: the Department of Natural and Territorial Sciences at the University of Sassari provided scientific expertise and knowledge in the field of environmental sciences.

  - University of Basilicata, School of Agricultural, Forestry, Food, and Environmental Sciences, Potenza: The University of Basilicata contributed with their expertise in the fields of agricultural, forestry, food, and environmental sciences.

Through the synergy among these institutions, it was possible to create the BC map of Sardinia, an important tool for understanding and studying the climates and biodiversity of the island [47]. The BC map represents the final stage of processing, achieved through the overlay of multiple layers such as Macrobioclimates, Phytoclimatic Plans, Ombrothermal Index, and Continentality Index. This intricate overlay generates a new layer that encompasses diverse combinations of bioclimatic values for each polygon. The resulting BC map comprises 43 classes of Isobioclimates, reflecting the detailed classification approach employed to capture the intricate characteristics of Sardinia's terrain. These 43 classes span a range of climate levels, encompassing dry, subhumid, humid, and hyperhumid conditions. Therefore, our focus lies on the subset of classes among the 43 available, specifically those that belong to one of the four predefined categories. By focusing on these, we generate a simplified map comprising two distinct macrogroups, categorized as follows: (1) "Dry", encompassing all the dry classes, and (2) "Humid", encompassing the remaining classes (subhumid, humid, and hyperhumid). The climate map carried out with this approach is reported on the right side of Figure 3.

By considering all the possible combinations between the two maps, six distinct classes are obtained, as illustrated in Figure 4. These classes are categorized as follows: dry-low, dry-medium, dry-high, humid-low, humid-medium, and humid-high. The integration of these distinct classes provides a comprehensive representation of the varying levels of dryness and density vegetation across the studied area. The resulting ancillary map is called the Biomass Dryness map (BD).

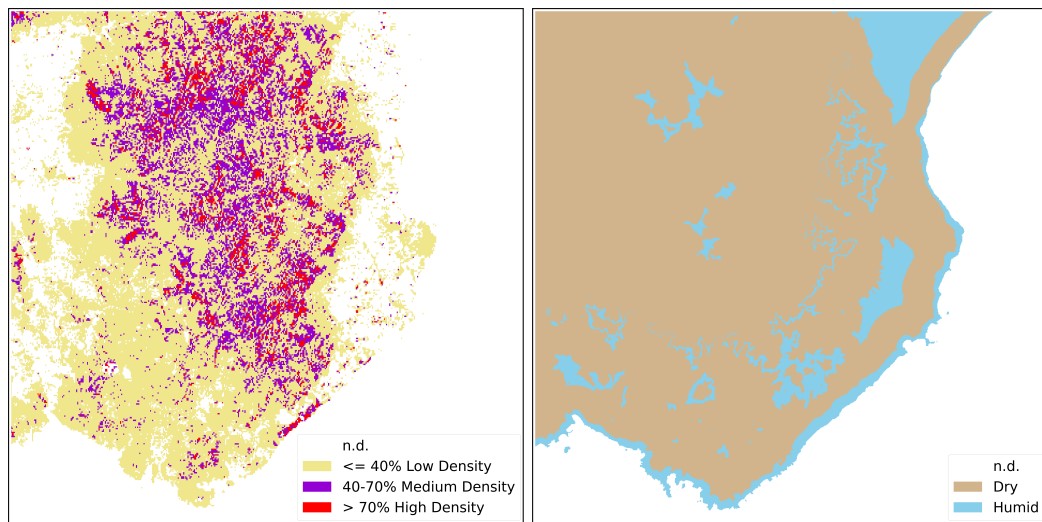

**Figure 3.** Ancillary maps: (on left side) the Above-Ground Biomass derived from ESACCI biomass map and (on the right side) the climatic zone map derived from the Sardinia Bioclimatic map; see text.

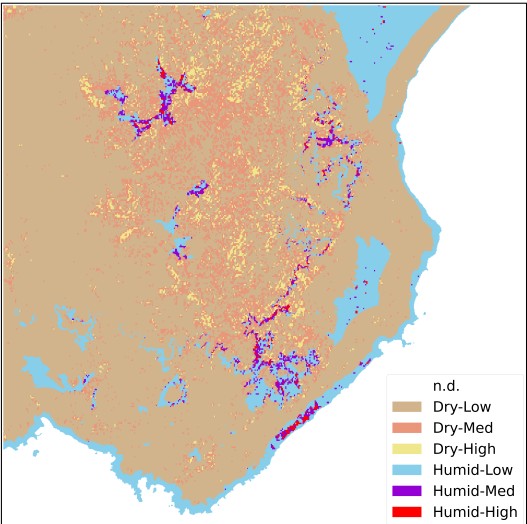

**Figure 4.** Combination of Above-Ground Biomass and Dryness Map into 6 classes. Where the terms: "Low", "Med" and "High" stand for low, medium and high density. The resulting ancillary map is called the Biomass Dryness map (BD).

### 2.4. Scott and Burgan Fuel Model

The standard fire behavior fuel models (SFBFMs) developed by Scott and Burgan [39] provide a systematic and standardized approach to characterizing vegetation and fuel properties, allowing fire analysts to assess and predict fire behavior under different fuel conditions. The SFBFM system serves as a critical tool in fire management decision-making processes. The SFBFM comprises a structured set of 45 distinct fuel models, out of which 5 are non-burnable types, namely Urban, Snow/Ice, Agricultural, Open Water, and Bare Ground. The remaining fuel models within the system are burnable and represent various vegetation and fuel characteristics. These models are carefully designed to capture the diverse range of fuel properties encountered in different fire-prone environments. The SFBFM system provides a comprehensive representation of fuel characteristics by considering factors such as fuel loadings, fuel particle sizes, and moisture content. Each fuel model is assigned a unique numeric code and accompanied by a descriptive name, facilitating standardized communication and data exchange among fire management professionals; a schematic description is reported in Table 3.

Table 3. Scott and Burgan fuel model scheme.

| Index | Description | Index | Description |
|---|---|---|---|
| GR1 | Short, Sparse, Dry Climate Grass | SH8 | High Load, Humid Climate Shrub |
| GR2 | Low Load, Dry Climate Grass | SH9 | Very High Load, Humid Climate Shrub |
| GR3 | Low Load, Very Coarse, Humid Climate Grass | TU1 | Low Load Dry Climate Timber–Grass–Shrub |
| GR4 | Moderate Load, Dry Climate Grass | TU2 | Moderate Load, Humid Climate Timber–Shrub |
| GR5 | Low Load, Humid Climate Grass | TU3 | Moderate Load, Humid Climate Timber–Grass–Shrub |
| GR6 | Moderate Load, Humid Climate Grass | TU4 | Dwarf Conifer With Understory |
| GR7 | High Load, Dry Climate Grass | TU5 | Very High Load, Dry Climate Timber–Shrub |
| GR8 | High Load, Very Coarse, Humid Climate Grass | TL1 | Low Load Compact Conifer Litter |
| GR9 | Very High Load, Humid Climate Grass | TL2 | Low Load Broadleaf Litter |
| GS1 | Low Load, Dry Climate Grass–Shrub | TL3 | Moderate Load Conifer Litter |
| GS2 | Moderate Load, Dry Climate Grass–Shrub | TL4 | Small Downed logs |
| GS3 | Moderate Load, Humid Climate Grass–Shrub | TL5 | High Load Conifer Litter |
| GS4 | High Load, Humid Climate Grass–Shrub | TL6 | Moderate Load Broadleaf Litter |
| SH1 | Low Load Dry Climate Shrub | TL7 | Large Downed Logs |
| SH2 | Moderate Load Dry Climate Shrub | TL8 | Long-Needle Litter |
| SH3 | Moderate Load, Humid Climate Shrub | TL9 | Very High Load Broadleaf Litter |
| SH4 | Low Load, Humid Climate Timber–Shrub | SB1 | Low Load Activity Fuel |
| SH5 | High Load, Dry Climate Shrub | SB2 | Moderate Load Activity Fuel or Low Load Blowdown |
| SH6 | Low Load, Humid Climate Shrub | SB3 | High Load Activity Fuel or Moderate Load Blowdown |
| SH7 | Very High Load, Dry Climate Shrub | SB4 | High Load Blowdown |

## 3. Method

The presented approach involves a two-step process. Firstly, a CNN-based classification is exploited to generate a primary land cover classification. This initial classification scheme provides a broad characterization of the different land cover classes (reported in Table 1) and exploits the Sentinel-2 multispectral images. Subsequently, the BD map is integrated to refine the primary classification adapting them according to the classification of fuel types defined by Scott and Burgan.

### 3.1. CNN Architecture

CNNs can be categorized into three main types: one-dimensional (1D), two-dimensional (2D), and three-dimensional (3D) CNNs. The numerical designation corresponds to the kernel's dimensions and complexity, which increase progressively. The 1D-CNN, while lacking spatial context, is particularly well suited for tasks involving the analysis of optical spectra or multitemporal data [48]. Therefore, the choice of 1D-CNNs for this specific application is driven by their simplicity and suitability for spectrum analysis.

The refinement of the model's design and hyperparameters involved an iterative process, including trial-and-error techniques. This systematic refinement aimed to strike a balance between complexity and efficiency while closely monitoring the results on the validation dataset. Specifically, the inclusion of two hidden layers yielded the best results in terms of accuracy and complexity. This choice represents a trade-off that offers the optimal accuracy improvement, as adding a third layer did not significantly enhance performance but substantially increased complexity. Conversely, using a single layer leads to lower performance. Therefore, employing two layers appeared to be the most suitable configuration for our specific case. The same principle applies to the selection of two fully

connected layers. After careful consideration, it was determined that incorporating two fully connected layers provided the best balance between model complexity and accuracy.

The primary objective of this systematic refinement was to achieve an optimal balance between model complexity and computational efficiency. Continuous monitoring of the validation dataset results was a key part of this process. This fine-tuning led to the derivation of the CNN structure depicted in Figure 5. The CNN's input consists of an array with a predetermined length of 16 elements, encompassing the 13 bands of the Sentinel-2 image as well as the 3 indices provided in Table 2. The first hidden layer of the model is a convolutional layer with 224 filters and kernel size 3. The output of this layer is passed through a max pooling layer that reduces the dimension of the feature map by a factor of 2. The second layer of the model is another convolutional layer with 112 filters and kernel size 3. Again, the output of this layer is passed through a max pooling layer that reduces the dimension of the feature map by a factor of 2. The pooled feature map is flattened in a vector of length 336 and passed through two fully connected layers. The first fully connected layer has 224 units, and the second fully connected layer consists of 128 units. Finally, the output is passed through a softmax activation function with 7 classes to produce the predicted class probabilities.

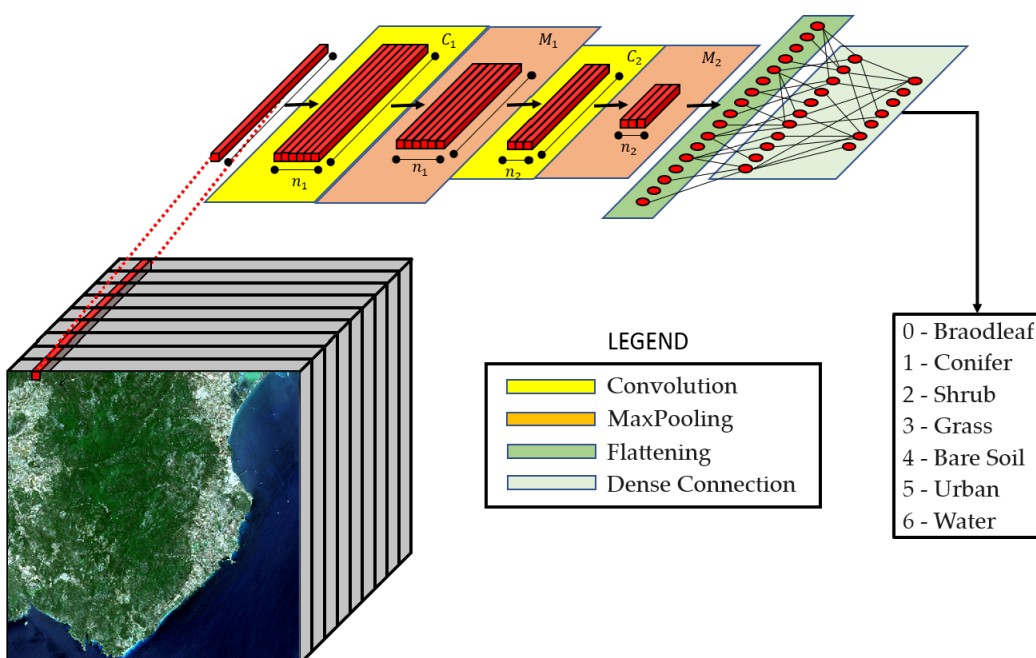

**Figure 5.** Scheme of proposed CNN.

After each convolutional or fully connected layer, the Rectified Linear Unit (ReLU) activation function is performed. This is a common so-called regularization technique that refers to techniques used to prevent overfitting, where the network becomes too specialized in the training data and performs poorly on new data. ReLU introduces non-linearity by returning zero for negative input values, which helps mitigate overfitting by promoting sparsity in neuron activation [33]. Another common regularization technique is dropout, which involves randomly deactivating a fraction of neurons during each training step, effectively reducing the network's reliance on specific neurons or connections [49]. By also applying dropout to the fully connected layers, the neural network becomes more resilient and better at handling unseen data, contributing to improved performance in various machine learning tasks. The model was trained using the Adam optimizer and the *categorical cross-entropy* loss function, with a maximum of 200 epochs and an automatic early stopping criterion based on a patience parameter of 30 epochs.

It is important to note that after performing the land cover classification using CNN, the classes Bare Soil, Urban, and Water are not taken into account when generating the fuel type map, as they fall directly into the non-burnable category. Therefore, from this point on, only the other classes will be considered in the methodology.

### 3.2. CNN-Based Unmixing

The final layer of the CNN employs a softmax activation function with seven classes, generating predicted class probabilities. This methodology exploits the inherent probabilistic features of CNN results to effectively deal with pixel confusion in scenarios where multiple classes coexist. Specifically, pixel confusion mainly results from pixels that include mixed features. For instance, when a pixel contains both grass and shrubs, the CNN output may exhibit a 50/50 balanced probability distribution. Importantly, this distribution reflects the presence of both classes rather than being an error in the CNN prediction. Leveraging this concept, multiple class assignments can be intelligently carried out from CNN prediction. In particular, the main objective is to identify the classes related to the GS and TU fuel models of the Scott and Burgan fuel model, specifically targeting the Grass–Shrub, Timber–Shrub, and Timber–Grass–Shrub classes (in this work, the term "Timber" includes both the Broadleaf and Coniferous classes).

In the practical implementation, following the prediction of the probability for each pixel within the area of interest, a subsequent pixel-level analysis is conducted. The following steps describe the algorithm that is performed for each pixel prediction:

1.   If the highest probability is above 60%, assign the pixel to the correspondent class.
2.   If the highest probability is below 60%, the second highest probability is above 20%, and the third highest probability is below 20%, assign the pixel to the two classes corresponding to the first two highest probabilities.
3.   If the highest probability is below 60%, and both the second and the third highest probabilities are above 20%, assign the pixel to the three classes corresponding to the first three highest probabilities.
4.   If the highest probability is below 60% and all other probabilities are below 20%, assign the pixel only to the highest probabilities.

The classes obtained after applying CNN-based classification and unmixing are as follows: Broadleaf, Conifer, Shrub, Grass, Grass–Shrub, Timber–Shrub (Broadleaf–Shrub and Conifer–Shrub), and Timber–Shrub–Grass (Broadleaf–Shrub–Grass and Conifer–Shrub-Grass).

### 3.3. CNN Test on Sardinia

While the training and validation phases were conducted using the dataset obtained through visual inspection (see Table 1), the testing process involved the entire image. During the testing process, the entire image was utilized, while predictions were compared and tested against external source maps, recognized as ground truth, to assess the model's reliability in classifying pixels beyond the original dataset. This was achieved by leveraging the four reference maps depicted in Figure 2.

### 3.4. Fuel Map Adaptation

To enhance the level of detail in the fuel map, a cross-referencing process was conducted between the land cover classification obtained through the previously described procedure and the BD map. This process resulted in a total of 42 classes, i.e., the initial 7 classes from the classification were combined with the 6 classes from the BD map. Subsequently, these 42 classes were associated with the corresponding fuel models from the Scott and Burgan system, as reported in Table 4.

**Table 4.** Adaptation of the fuel types to the Scott and Burgan system, with the symbols BL, CF, SH, GR, GS, TS, and TSG representing Broadleaf, Conifer, Shrub, Grass, Grass–Shrub, Timber–Shrub, and Timber–Shrub–Grass classes, respectively.

|       | Dry-Low | Dry-Med | Dry-High | Hum-Low | Hum-Med | Hum-High |
|-------|---------|---------|----------|---------|---------|----------|
| **BL**  | TL2 | TL6 | TL9 | TL2 | TL6 | TL9 |
| **CF**  | TL1 | TL3 | TL5 | TL1 | TL3 | TL5 |
| **SH**  | SH2 | SH5 | SH7 | SH6 | SH3 | SH9 |
| **GR**  | GR2 | GR4 | GR7 | GR5 | GR6 | GR9 |
| **GS**  | GS1 | GS2 | SH7 | GS3 | GS4 | SH8 |
| **TS**  | TU1 | TU1 | TU5 | SH4 | TU2 | TU2 |
| **TSG** | TU1 | TU1 | TU5 | SH4 | TU3 | TU3 |

## 4. Results

This section presents the performances of the CNN model in terms of training accuracy, test accuracy, and comparisons with RF and SVM, as well as the final classification of the fuel map aligned with the Scott and Burgan fuel model.

### 4.1. Performances of CNN Classification

The training and validating phases utilized the dataset derived from visual inspection, while the test involved the entire image and cross-referencing with external source maps as ground truth, ensuring a robust assessment of the model's performance. Moreover, a comparison is conducted with RF and SVM to underscore the potential of the CNN-based fuel map generator.

#### 4.1.1. Model Accuracy

In the practical implementation, the dataset reported in Table 1 was divided into train and validation subsets, with split ratios of 70% and 30%, respectively. The CNN model was implemented in Python using TensorFlow [50] on a personal computer equipped with 12 GB of RAM, an Intel Core i7-12700H 2.70 GHz CPU from the 12th Generation, and an NVIDIA GeForce RTX 3080 Ti GPU with 16 GB of dedicated RAM (manufactured by Intel and NVIDIA Corporation, Santa Clara, CA, USA). The results from the CNN model's training, with precision (0.99), recall (0.99), and F1 (0.99) scores, indicate an exceptionally high level of performance. This demonstrates that the network has been trained effectively to accurately identify pixels. In Figure 6, one can appreciate the segmentation results over the region of interest on the left side, and the confusion matrix on the right side. One can notice that the Not Burnable classes (Bare Soil, Urban, and Water) are perfectly recognized, while a slight degree of confusion is observed among the other classes due to increasing spectral similarity.

#### 4.1.2. CNN Test and Comparative Analysis with RF and SVM

The CNN's segmentation of the entire area of interest (shown on the left side of Figure 6) is cross-referenced with the four ground truth maps (shown in Figure 2) to evaluate the CNN model's ability to accurately classify previously unseen pixels. The same evaluation is applied to both the RF and SVM models for the purpose of comparing the performances. This method enabled a thorough evaluation of the CNN's performance across diverse class categories, offering valuable insights into its generalization capabilities and adaptability on previously unobserved pixels.

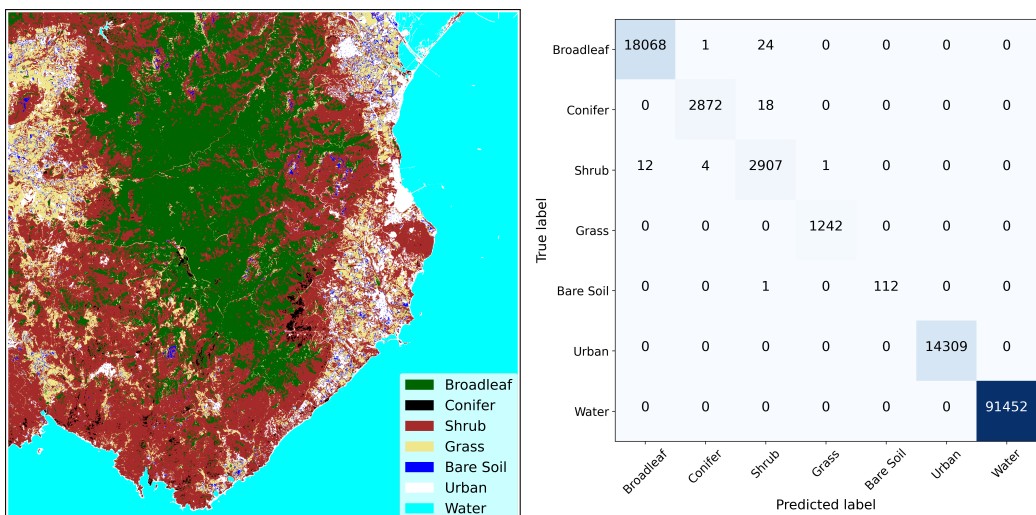

**Figure 6.** Segmentation and confusion matrix results.

The RF model was implemented using the "RandomForestRegressor" function from the "catboost" library in Python [51]. The model was set with 500 boosting iterations and a maximum tree depth of five to avoid overfitting. The 'MultiRMSE' loss function was employed for optimization, and a learning rate of 0.1 was utilized to control the step size during training. The computations were performed on a GPU, and early stopping with 500 rounds was applied to prevent overfitting.

The SVM model was implemented using the "SVC" function from the "scikit-learn" library in Python [52]. For the classification task, the radial basis function ("rbf") was chosen as the kernel function. To ensure repeatability, a random seed of 0 ($randomstate = 0$) was set. Additionally, the model was configured to provide class probabilities using the probability parameter set to True. A gamma value of 100 was employed for model tuning, which influences the flexibility of the decision boundary.

The results presented in Table 5 show the training performances of all three models in terms of accuracy, recall, and F1 scores. The results suggest that all three models exhibit similar training performances, showing only minor variations.

**Table 5.** Comparison in terms of accuracy, recall, and F1 scores.

|  | Accuracy | Recall | F1 Score |
| --- | --- | --- | --- |
| **CNN** | 0.99% | 0.99% | 0.99% |
| **RF** | 0.99% | 0.99% | 0.98% |
| **SVM** | 0.99% | 0.98% | 0.98% |

In Table 6, the test accuracy of each of the Burnable classes (Broadleaf, Conifer, Shrub, and Grass) is reported for all three models. The results, once again, demonstrate relatively similar performances among the models. Notably, all of the models achieve the impressive accuracy of 99% for the Broadleaf class, indicating exceptional ability to accurately classify the majority of pixels belonging to this category. However, the CNN model demonstrates a notable advantage, particularly in the classification of Coniferous land cover. The CNN model achieves an accuracy of 0.78%, while RF and SVM achieve accuracies of 0.70% and 0.60%, respectively.

Similar observations can be made for the Grass class, although the differences are less pronounced. In this case, the CNN model outperforms the others with an accuracy of 84%, while RF and SVM achieve accuracies of 81% and 79%, respectively.

Thus, despite the similarities in overall performance, the advantage of the CNN model in classifying Coniferous and Grass regions highlights its potential as a more adept solution for tasks that require discerning subtle and complex patterns in land cover data. These

findings further emphasize the CNN's proficiency in image analysis and applicability in remote sensing.

**Table 6.** Comparison in terms of accuracy for Broadleaf, Conifer, Shrub and Grass classes.

|  | **Broadleaf** | **Conifer** | **Shrub** | **Grass** |
| --- | --- | --- | --- | --- |
| **CNN** | 0.99% | 0.79% | 0.76% | 0.84% |
| **RF** | 0.99% | 0.70% | 0.77% | 0.81% |
| **SVM** | 0.99% | 0.60% | 0.78% | 0.79% |

*4.2. Fuel Map Generation*

By utilizing a CNN-based classification approach with Sentinel-2 imagery, the fuel types are precisely categorized into seven primary classes. Subsequently, subclasses are derived by integrating the BD map, which effectively aligns the classification with the Scott and Burgan fuel system. Considering the high accuracy observed in the proposed classification, the resulting fuel map is expected to exhibit similar levels of accuracy. This map, reported in Figure 7, is a product of the integration and cross-referencing of these complementary maps, ensuring a robust and reliable classification outcome.

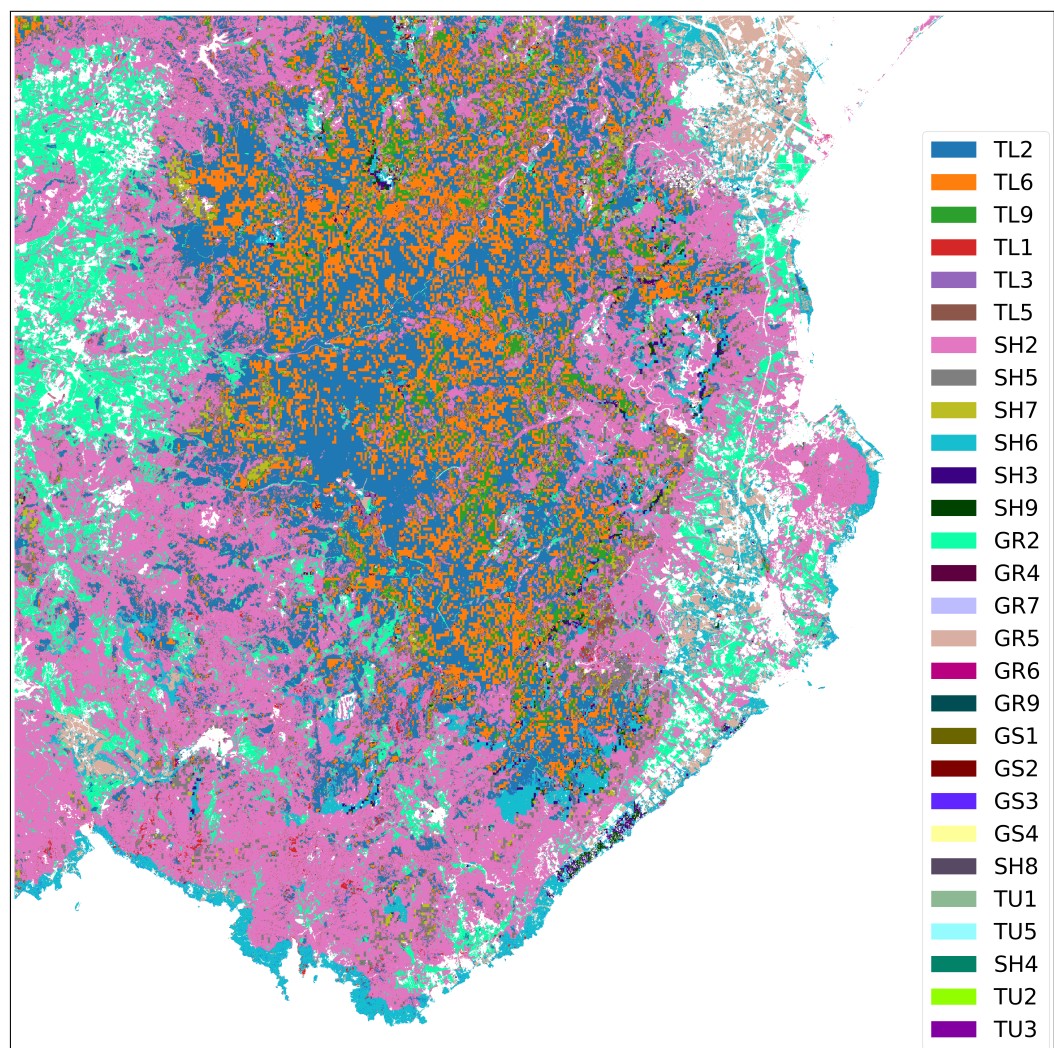

**Figure 7.** Fuel map adaptation to Scott and Burgan.

## 5. Discussion

This research was driven by the need to advance land cover classification methodologies, fuel mapping techniques, and remote sensing applications. In response to this demand, the research introduces a CNN-based fuel map generator utilizing Sentinel-2A satellite images.

The presented results provide valuable insights into the performance and potential applications of the CNN model for land cover classification and fuel map generation. The CNN model demonstrates exceptional accuracy (0.99%), recall (0.99%), and F1 scores (0.99%), indicating its proficiency in accurately identifying pixels. One can notice that the Not Burnable classes (Bare Soil, Urban, and Water) are accurately recognized by the model, as these classes possess distinct spectral patterns that facilitate reliable identification. However, there is a slight degree of confusion observed among the other classes, mainly attributed to the increasing spectral similarity, which makes accurate classification more challenging. It is worth mentioning an intriguing observation, although somewhat unrelated to the main focus, where the model successfully classifies some vessels located off the East Coast as "urban", even when they were barely visible to the naked eye. This demonstration of the model's overall capability and accuracy is noteworthy and piques further interest in its potential applications. This ability is also exhibited by the RF and SVM classifiers, highlighting the robustness and potential of NN models in advancing land cover mapping.

In the testing phase, the CNN model's robustness is evaluated by comparing its predictions against external source maps as ground truth. The testing results demonstrate high accuracy, particularly in distinguishing the Broadleaf (0.99%) and Grass (0.84%) classes. The model's ability to accurately classify these classes can significantly contribute to land cover mapping and monitoring applications. However, the slightly lower accuracies for the Shrub (0.76%) and Conifer (0.78%) classes suggest that further refinement or data augmentation may be beneficial for improving their performance. It is worth noting that the variations in accuracy across classes could be due to the inherent differences in features and characteristics among these land cover categories. Additionally, other factors, such as data imbalance, the availability of training samples, and class complexity, might influence the results. Overall, these results suggest that the CNN model performs well in classifying land cover types, with particularly outstanding performance for the Broadleaf class. These high validation accuracy values provide evidence of the model's ability to generalize well beyond the original training dataset. This generalization capability is crucial for real-world applications, where the model is expected to accurately classify land cover in diverse and previously unseen regions.

The comparison with the RF and SVM models reveals that all three models exhibit similar performances with minor variations. In particular, the CNN model shows a slight advantage in classifying Coniferous land cover, which underscores its potential in discerning subtle and complex patterns in land cover data. The CNN achieves an impressive accuracy of 0.78 for Coniferous regions, while the RF model falls short with an accuracy of 0.70 for the same category.

The fuel map generation process provides a comprehensive representation of fuel characteristics by aligning the land cover classification with the well-known Scott and Burgan fuel system, effectively categorizing fuel types into seven primary classes and derived subclasses. This detailed fuel map exhibits high accuracy, which is crucial for fire risk assessment and management efforts. This accurate and detailed fuel map serves as a valuable resource for decision-makers involved in land management, fire risk assessment, and various other domains. It empowers authorities, stakeholders, and land managers to make well-informed decisions regarding wildfire prevention strategies, resource allocation, and emergency response planning. Moreover, the fuel map's applications extend beyond fire management. It supports ecosystem restoration, habitat conservation, and resource management initiatives, highlighting its role in enhancing wildfire management and promoting ecological resilience.

Furthermore, in contrast to previous methods, this study introduces significant advancements. Firstly, it generates a high-resolution map with a granularity of 10 m, a substantial improvement compared to the 1-kilometer resolution in [38]. Moreover, it streamlines the process by relying on readily available and relatively stable external maps, ABG and BC, thereby reducing the need for extensive data acquisition campaigns. Additionally, in contrast to the approach detailed in [36], the results of the proposed method seamlessly integrate with the established fuel model classification by Scott and Burgan, making it a valuable, efficient, and accurate tool for generating fuel maps.

The method's limitation could be its reliance on external biomass and climate maps as fundamental inputs for generating the final fuel map classification. Moreover, the availability of such maps may be limited across different regions. The availability of these maps may vary across different regions and could pose challenges for areas where such data are scarce or unavailable. To address this, future research could focus on developing an autonomous AGB estimation system solely based on Sentinel-2A spectral images, eliminating the need for external maps. However, this poses a complex challenge, requiring a deep understanding of spectral signals and precise model calibration and validation with ground reference data. Thus, the AGB map provided by ESA can serve as the ground reference label for training the CNN. This method could offer promising opportunities but also faces some potential limitations. For instance, the accuracy of the ESA AGB map itself is crucial, such as the spatial and temporal alignment of ESA's AGBs with Sentinel-2A images, as any errors or uncertainties in this map can directly impact the reliability of the CNN's predictions. Moreover, the limited coverage of the ESA AGB map may pose challenges, particularly in remote or inaccessible areas, where biomass estimates may be unavailable. It is essential to ensure a comprehensive and representative training dataset to account for different land cover types and biomass levels to avoid biases in the model.

In conclusion, while the proposed CNN-based method using the ESA AGB map as a label offers exciting possibilities for autonomous fuel mapping, it is crucial to address the potential limitations mentioned above to ensure the method's accuracy and practicality in real-world applications. By mitigating these challenges, the proposed approach can make valuable contributions to land cover classification and biomass estimation, supporting various fields such as environmental management, disaster response, and resource conservation.

## 6. Conclusions

In conclusion, recent years have demonstrated the fundamental role of machine learning in harnessing the enormous volume of data generated daily by satellite remote sensing. The motivation for this paper stems from the research conducted within the FirEUrisk project, driven by the desire to both solidify the foundations of these novel techniques and provide a functional open-source tool for fuel type mapping. This work has shed light on how the primary machine learning approaches are indispensable tools in this context. Notably, the CNN's superior performance, especially in distinguishing between the Broadleaf and Conifer classes, underscores its potential for land cover mapping and monitoring tasks. Furthermore, aligning our fuel type classification with the widely recognized Scott and Burgan (2005) standard is a pivotal step in harmonizing our efforts with established wildfire management protocols. This integration ensures compatibility and simplifies the adoption of this innovative approach into current fire management practices. This tool paves the way for a more efficient and automated analysis of large-scale satellite imagery, benefiting various fields such as agriculture, forestry, disaster monitoring, and environmental studies. The results of this study could be expanded in future works that consider both bigger and more geographically diverse training datasets and more complex neural networks capable of carrying out an ABG prediction.

**Author Contributions:** Conceptualization, A.C. and G.L.; methodology, A.C. and G.L.; software, A.C. and D.S.; validation, A.C.; formal analysis, A.C.; investigation, A.C.; resources, G.L.; data curation, A.C.; writing—original draft preparation, A.C.; writing—review and editing, A.C., D.S. and G.L.; visualization, A.C.; supervision, D.S. and G.L. All authors have read and agreed to the published version of the manuscript.

**Funding:** This research received no external funding.

**Institutional Review Board Statement:** Not applicable.

**Informed Consent Statement:** Not applicable.

**Acknowledgments:** The subject of this paper has been motivated by the research activity carried out in the framework of the FirEUrisk project.

**Conflicts of Interest:** The authors declare no conflict of interest.

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
