# Peer review of "Fuel Type Mapping Using a CNN-Based Remote Sensing Approach: A Case Study in Sardinia"

_fire, doi:10.3390/fire6100395_

Round 1
Reviewer 1 Report
The authors presented a method to obtain a map of fuel load aligned with the SFBFM of Scott and Burgan over the fire-prone island of Sardinia in Italy, using three datasets: a landcover dataset from Sentinel-2 images classified using a Convolutional Neural Network algorithm, an Above Ground Biomass map from ESACCI, and a Bioclimatic map of the region developed regional institutions and universities.
The work is interesting, written clearly and well-structured. I have highlighted one methodological issue that should be addressed by the authors before publication.
MAJOR:
I have one major comment regarding the training and testing approach, that I will separate in three main points:
1. The authors carefully selected the most pure pixels in order to well-discriminate the classes. My question is whether this approach is supported in the literature as being the best way to obtain better results? The challenge in classification is to correctly discriminate the less pure pixels. Why not train a classification algorithm using also less pure pixels?
2. The accuracy results are extraordinarily high. However this raise concerns about the selection of training/test pixels which, as mentioned before, are the purest, which we may say, the “easiest to be correctly recognized by the algorithm. By using this approach, we don’t have any information on the capabilities of this algorithm in discriminating the classes in more complex landscapes. This is a shortcoming of this approach that should be addressed or at least acknowledged in the manuscript.
3. Section 4.1.2.: This looks to me a cross-comparison with other landcover datasets. These products are also subject to the same omission and commission errors. It doesn’t seem wise to use them as reference to evaluate the performance of your product. Your product may probably be better than these other product that cover larger areas (global, regional). The metrics that are derived from the comparison may be referred as “consensus” between products. It is actually more interesting to investigate where and why your product performs better or worse than others. Maybe beyond the scope of this manuscript, but an idea would be to use pixels from visual interpretation such as in the training/test phase to cross-validate all these products.
Moreover, a table summarizing this cross-comparison would be of value.
Another major remark is about some self-citations that are unnecessary: Under the same citation point in the manuscript (line 64, references 23-29), the authors list 7 works where CNN where used in the field of wildfire detection, 5 of which are from one of the authors. This is inappropriate because there are many other works from other authors using these techniques. Moreover, reference 24 doesn’t use CNN and the 5 works from the author are very similar to each other (hyperspectral imagery and CNN).
MINOR:
Line 34-35: change “develop and evaluate” to “developed and evaluated”.
Line 36-37: “passive and active Sentinel data” may be clarified by specifying that it’s the sensor to be active or passive. For example: “The combination of data collected using passive and active sensor from the Sentinel family of satellites…”. Also is that sentence related to the previous one or is it the result of another published work? Please, adjust accordingly.
Line 41: why reference 16 is indicated differently?
Line 54: change “develops” to “developed”
Line 55: correct “exploited”
Line 57: change “providing” to “reaching” and correct “work”
Line 79: Please provide a reference for the Scott/Burgan fuel classification system, even if it is described and referenced properly in section 2.4
Section 2.2.: Please provide a short description of the main 6 classes in the study area. In particular, because you reported the difficulty of selection of pure pixels for the Grass and Bare Ground classes, makes the reader wondering what type of landscapes belong to these classes (beaches, rocky surfaces, agriculture land…)
Line 154: correct “aligned”
Line 176: change “reported” with “shown”
Line 219: BD is the one in Fig. 3? If yes, then indicate in the Figure 3 caption
Section 3.1.: All the technical part of the construction of the CNN is lacking references, full names indicated by acronyms, and short explanations on the reasons of such choices (why relu activation function, why 2 hidden layers, etc.).
Line 303-310: This kind of sentences are not directly related to the Methodology section. They may be more suitable for the Discussion section.
Line 334-352: The description of the validation methodology should be moved to the “Methodology” section.
Section 4.2. and Fig 6.: Why you do not report the final fuel map of the whole study region?
Line 411-413: Is this that exceptional? Every classifier should be able to do that given the spatial resolution of the images. Have you checked if RF and SVM were equally able to discriminate the boats?
Line 453: A “be” may be missing after “could”
Line 463: correct with “could offer” or “offers”
Line 490: fix “diver”
English is good. Just a few edits that I mentioned in my comments.
Author Response
Thank you very much for dedicating your time to review our manuscript. We have provided comprehensive responses in the attached document, along with corresponding revisions and corrections that are highlighted in the resubmitted files.

Reviewer 2 Report
In this work, a method for discriminating between wildfire fuel types by exploiting remote sensing data and CNN was presented. This method can be utilized by fire management agencies, policymakers, and researchers for improved fire behavior prediction and mitigation practices. This contributes to enhancing the accuracy and effectiveness of fire management efforts. However, the following problems need to be modified:
1. There are multiple errors in the language, for example, the “exploiteed” in line 55, “descrition” in line 147, “aligbned” in line 154. They're all spelling mistakes; The “false RGB” in line 130 should be written as “Pseudo-color”; The use of abbreviations is also irregular, for example, in line243 both abbreviations and full names exist simultaneously. The above examples do not contain all errors. Please check the manuscript carefully and correct the errors.
2. The amount of work in this study is relatively small, and a lot of space is spent in the manuscript to emphasize the role of CNN technology applied to fuel maps, and the explanation of technical advantages is less. It is suggested that the author can reflect the technical advantages of CNN model through the recognition and training of satellite maps in different seasons.
3. The manuscript's sentences are not sufficiently refined and contain too many modifiers. Many contents are repeated several times, for example, CNN's recognition rate of conifer trees is repeated three times.
4. The font in the figure should be consistent with the text.
5. How are the formulas for each index in Table 1 obtained? Need to explain or reference. What is the basis for probability distribution in the algorithm performed by pixel prediction in section 3.2, and which three classes are referred to in the algorithm?
6.line 279 mentions unifying broadleaf and coniferous into timber. However, it can be found from Table 2 that The number of labelled pixels of the two are very different. Is this approach reasonable?
7. It can be seen from Table 2 that the number of “urban” is much greater than that of “shrubs”, but the result of Figure 5 is quite opposite. Why is this?
8. The manuscript explains that the performance of the SVM model is slightly worse than that of the RF model, and Table 5 shows their differences. This conclusion is contrary to the conclusion in reference 18. Why? Moreover, they only have a 1% gap in Recall, so the precision results of RF model can also be presented in Table 6. After all, there is only a 1% gap between CNN model and RF model in F1-Score.
9.line439-440. The manuscript does not show CNN's strengths in geospatial research
The manuscript needs small revision for language and grammar
Author Response

(The authors gave the same response as above.)

Reviewer 3 Report
In general, the manuscript is interesting and relevant to the journal but needs major revisions to be published in this Journal. More and detailed comments are provided in the manuscript file.

In general, the manuscript is interesting and relevant to the journal but needs major revisions to be published in this Journal. More and detailed comments are provided in the manuscript file.
Author Response
Thank you very much for dedicating your time to review our manuscript. We have provided responses in the attached document, along with corresponding revisions and corrections that are highlighted in the resubmitted files.

Round 2
Reviewer 1 Report
The authors have provided answers to my questions and modified the manuscript accordingly. Although I disagree with some aspects of their reasoning, their motivation are sufficiently reported in the manuscript.
I'll let the language editing team deal with minor spell checks
Reviewer 2 Report
The authors have made detailed revisions based on the suggested comments and the current manuscript is acceptable.
The sentences in the manuscript are not refined enough. Minor editing required for English.
Reviewer 3 Report
The manuscript has been revised according to the suggestions and comments of the reviewers